# Maternal Adherence to the Mediterranean Diet during Pregnancy: A Review of Commonly Used *a priori* Indexes

**DOI:** 10.3390/nu13020582

**Published:** 2021-02-10

**Authors:** Marion R. Eckl, Elske M. Brouwer-Brolsma, Leanne K. Küpers

**Affiliations:** 1Division of Human Nutrition and Health, Wageningen University & Research, 6708 WE Wageningen, The Netherlands; marion.eckl@wur.nl; 2The Generation R Study Group, Erasmus MC, University Medical Center Rotterdam, 3000 CA Rotterdam, The Netherlands; leannekupers@gmail.com

**Keywords:** Mediterranean diet, nutrition assessment, *a priori* index, pregnancy, maternal nutrition

## Abstract

Currently, many *a priori* indexes are being used to assess maternal adherence to the Mediterranean diet (MD) during pregnancy but each with different components, cut-off points, and scoring systems. This narrative review aimed to identify all observational studies utilizing *a priori* indexes to assess maternal adherence to the MD during pregnancy. A systematic search was conducted in Pubmed until 1 July 2020. Among the 27 studies included, eight different *a priori* indexes were identified. Studies included a range of 5 to 13 dietary components in their indexes. Only three dietary components—vegetables, fruits, and fish—were common among all indexes. Dairy and alcohol were the only two components modified for pregnancy. All but one study either excluded alcohol from their index or reversed its scoring to contribute to decreased adherence to the MD. Approximately half of the studies established cut-off points based on the distribution of the study population; the others utilized fixed criteria. This review emphasizes the incongruent definitions of the MD impairing effective comparison among studies relating to maternal or offspring health outcomes. Future research should carefully consider the heterogeneous definitions of the MD in *a priori* indexes and the relevance of incorporating pregnancy-specific nutritional requirements.

## 1. Introduction

The Mediterranean diet (MD) has long been heralded as one of the healthiest dietary patterns worldwide [1]. Traditionally, the MD reflects the common dietary characteristics of populations native to the Mediterranean basin, including countries such as Greece and Italy [2]. The predominant elements of the MD consist of high intakes of vegetables, fruits, legumes, nuts, seeds, whole grains, and olive oil; moderate intakes of fish and alcohol; low to moderate intakes of dairy; and low intakes of meat and poultry [3]. As such, the MD is largely plant-based abundant in fiber and antioxidants as well as monounsaturated (MUFA) and polyunsaturated fatty acids [2,4]. Over the past decades, numerous epidemiological studies have investigated the influence of adherence to the MD on health outcomes, ultimately showing protective associations with a host of non-communicable diseases ranging from cardiovascular disease (CVD) to cancer [5]. Recently, systematic reviews have also reported protective associations between maternal adherence to the MD during pregnancy and various maternal and offspring outcomes, including gestational diabetes mellitus (GDM) [6] and offspring adiposity [7,8]. 

To date, several *a priori* indexes have been developed to assess adherence to the MD using a scoring system based on predefined features of the MD [9,10,11]. In fact, a narrative review identified 22 different indexes used to assess adherence to the MD in adult and elderly populations [10]. In these indexes, beneficial foods contribute to increased adherence to the MD, whereas detrimental foods contribute to decreased adherence to the MD. Yet, each index features vastly different components, cut-off points, and scoring systems [10]. To illustrate, the original Mediterranean diet score (MDS) was developed by Trichopoulou et al. (1995) and consists of six beneficial components (i.e., vegetables, fruits and nuts, cereals, legumes, alcohol, and a ratio of MUFA to saturated fatty acids (MUFA:SFA)) as well as two detrimental components (i.e., meat and dairy), each of which are scored 0–1 based on the median intake levels of the study population [12]. In 2003, the original MDS was modified by the same authors to produce the MDS-2003, which additionally include fish as a beneficial component [13]. Conversely, the MDS-2006 utilizes eleven components, adding separate categories for potatoes and poultry and replacing MUFA:SFA with olive oil, each of which are scored 0–5 based on fixed servings of consumption [14]. Alternatively, the Prevención con Dieta MEDiterránea (PREDIMED) score uses the Mediterranean Diet Adherence Screener (MEDAS), a 14-item questionnaire consisting of twelve food frequency (e.g., daily amount of fruit consumed) and two food habit questions (e.g., use of olive oil during cooking), for rapid assessment of adherence to the MD during trials or clinical settings [15]. Although the PREDIMED score includes many components traditional to the original MDS and the MDS-2003 [12,13], it also incorporates components non-traditional to these indexes, including sweetened or carbonated beverages and commercial pastries [15]. 

Beyond the inherent differences among *a priori* indexes, nutritional requirements during pregnancy further complicate the application of *a priori* indexes in pregnant populations [16]. Although moderate alcohol consumption is considered a beneficial component in the traditional MD [2,4], abstinence from alcohol is widely recommended during pregnancy to avoid adverse fetal development [17,18]. Furthermore, several micronutrient requirements (e.g., folic acid, iron, and calcium) increase during pregnancy [19]. Consequently, many epidemiological studies evaluating maternal adherence to the MD during pregnancy on maternal and offspring health outcomes have altered the original designs of *a priori* indexes in order to address these nutritional requirements during pregnancy [20,21,22,23]. Such modifications include scoring alcohol as a detrimental component [22] or removing it entirely from the index [20,21,23] as well as scoring dairy as a beneficial component to meet increased calcium requirements [20]. In 2009, Mariscal-Arcas et al. (2009) proposed an *a priori* index specifically for use during pregnancy called the MDS for pregnancy, which incorporated folic acid, iron, and calcium but maintained dairy as a detrimental component in the scoring system [16]. 

Whether due to inherent differences or subsequent modifications in light of pregnancy, the heterogeneity in the inclusion and discrimination of components in *a priori* indexes greatly challenges comparison among studies [9,11], thus, impeding the ability to definitively draw conclusions on maternal and offspring health outcomes. Therefore, the goal of this narrative review was to identify all observational studies utilizing *a priori* indexes to assess maternal adherence to the MD during pregnancy with a particular emphasis on evaluating the food and nutrient components, namely the choice and discrimination of dietary components in these indexes, in addition to the cut-off values and scoring systems. 

## 2. Materials and Methods 

A comprehensive search was conducted in Pubmed in order to identify all observational studies published evaluating maternal adherence to the MD during pregnancy without time limits through 1 July 2020. The search string comprised keywords and Medical Subject Headings (MESH) terms relating to maternal dietary intake during pregnancy and adherence to the MD as described in Table 1. The initial database search was also supplemented by a manual search of reference lists of relevant studies in order to identify studies not retrieved by the initial search in Pubmed. Only observational studies (i.e., cohort studies, case-control studies, and cross-sectional studies) that utilized an *a priori* index to assess the exposure of maternal adherence to the MD during pregnancy were included in this review. Reasons for excluding articles from the review included: (1) Irrelevant population, (2) irrelevant exposure, (3) incomplete information on the *a priori* index in methods, (4) non-observational study design, (5) review, meta-analyses, editorials, or conference proceedings, and (6) no English translation available. Figure 1 describes the literature search and selection process in more detail. From the studies selected for inclusion in this review, we extracted data on the authors and year of publication; study design; population characteristics; method of dietary assessment; the definition, cut-off values, and scoring system of the *a priori* index; as well as the main outcomes.

## 3. Results

### 3.1. Study Characteristics

In total, 27 observational studies were included in the review: 17 (63.0%) cohort studies [20,21,22,24,25,26,27,28,29,30,31,32,33,34,35,36,37]; 7 (25.9%) cross-sectional studies [38,39,40,41,42,43,44]; 2 (7.4%) case-control studies [23,45]; and, 1 (3.7%) nested case-control study [46]. Table 2 provides a summary of the key characteristics of these 27 studies. The studies were published from 2008 to 2020. 20 studies were conducted in Europe of which 17 were conducted in the Mediterranean countries of Spain [21,22,23,25,26,27,28,29,30,40,41,42,44,46], Greece [20,28,29,38], and Italy [43], whereas the other three were conducted in Norway [31], Denmark [35], and the United Kingdom [24]. Seven studies were conducted in North America including the United States [20,32,33,34,36], Mexico [39], and the Caribbean [37]. One study was conducted in Iran [45]. Notably, three studies utilized cohorts from two geographical locations in their studies, including Greece and Spain [28,29] and Greece and the USA [20].

In terms of outcomes, 23 studies evaluated associations between exposure of maternal adherence to the MD and offspring health outcomes: Including newborn anthropometrics and risk of preterm birth (*n* = 10) [23,28,31,35,36,37,38,42,44,46]; risk of atopic diseases (*n* = 7) [24,25,26,27,29,32,39]; and the development of cardiometabolic disorder risk factors (*n* = 6) [20,21,30,34,40,41]. Three studies assessed maternal adherence to the MD in relation to maternal health outcomes: Including the risk of GDM (*n* = 1) [45] and levels of homeostasis model assessment of insulin resistance and adiponectin (*n* = 2) [33,43]. One study investigated the change in adherence to the MD across trimesters of pregnancy [22].

### 3.2. Type of Index

Eight a priori indexes were identified among the included studies for the assessment of maternal adherence to the MD during pregnancy. The most commonly used index was the MDS-2003 (*n* = 12) [20,23,24,27,28,29,32,33,34,37,39,45] followed by the PREDIMED score (*n* = 5) [23,40,41,43,44] and the relative MDS (rMed) (*n* = 3) [21,22,30]—a variation of the original MDS [12] and MDS-2003 [13] developed in a Spanish cohort [49]. Castro-Rodriguez et al. (2010) and (2016) reported using a modified version of the MDS-2004 by Psaltopoulou et al. (2004) [48] but with additional food components (i.e., potatoes, pasta, rice, and fast food) included at the discretion of the study group [25,26]. Others used the MDS-2006 (*n* = 2) [23,38]; a modified version of the Mediterranean diet quality index in children and adolescents (KIDMED) index (*n* = 2) [42,46]; as well as Khoury’s criteria (*n* = 2) [31,35] based on principles established in a randomized controlled trial on pregnant women [51]. Only one study used the alternative MD (aMed)—another variation of the MDS-2003 [59] adapted in a US cohort—in their main analysis [36]. Almost all studies used one index in their assessments of maternal adherence to the MD during pregnancy with the exception of Martínez-Galiano et al. (2018) in which three indexes were utilized: The MDS-2003, PREDIMED score, and MDS-2006 [23]. Notably, Fernández-Barrés et al. (2016) reported using rMed in the main analysis and aMed in the sensitivity analysis. However, no information was provided in the methods on the specific components, cut-off values, and scoring system of aMed; for this reason, only their use of rMed was evaluated in this review [21].

### 3.3. Dietary Assessment Methods

A total of 22 studies employed a food frequency questionnaire (FFQ) to assess maternal dietary intake during pregnancy [20,21,22,23,24,25,26,27,28,29,30,31,32,34,35,36,37,38,39,40,41,43] of which 14 studies measured dietary intake once [20,23,24,25,26,27,28,29,31,35,36,37,38,39] and 7 studies measured dietary intake twice during pregnancy [21,30,32,34,40,41,43]. Jardí et al. (2019) was the only study using an FFQ to measure dietary intake three times during each trimester of pregnancy [22]. Of the five studies that calculated the PREDIMED score [23,40,41,43,44], four studies reported using an FFQ to assess dietary intake [23,40,41,43], which was then presumably grouped into the components of the MEDAS questionnaire in order to produce the PREDIMED score [15]. Contrastingly, Tomaino et al. (2020) reported not using an FFQ, but rather only the MEDAS questionnaire, to both assess dietary intake during pregnancy and to produce the PREDIMED score [44]. Similarly, the two studies utilizing the modified KIDMED index reported using only the KIDMED questionnaire [42,46], a 16-question test designed based on the principles of the MD, rather than an FFQ [55]. Furthermore, two studies assessed dietary intake by means of repeated 24-h recalls [33,45]. One study measured dietary intake once at an unspecified time period during pregnancy with three 24-h recalls [45]; and the other study measured dietary intake three times during early, mid-, and late pregnancy each with three 24-h recalls [33].

### 3.4. Food and Nutrient Components

Overall, the number of dietary components included in the indexes ranged from 5 [31,35] to 13 [42,44,46] depending upon the chosen index and any subsequent modifications to that index. However, in total, 17 different dietary components were identified among the indexes of the twenty-seven studies. 10 components were considered traditional to the components included in the original MDS and MDS-2003 [12,13]; seven components were considered non-traditional to these original indexes [12,13]. Beneficial components are defined as dietary components contributing to increased adherence to the MD, whereas detrimental components are defined as dietary components contributing to decreased adherence to the MD. Table 3 provides a comparison of the food and nutrient components incorporated into the *a priori* indexes of the 27 included studies. Of note, Martínez-Galiano et al. (2018) utilizes three different *a priori* indexes; therefore, the number of studies does not always add up to 27 when the inclusion, discrimination, or scoring of components differed among the three indexes. Any discrepancies among the three indexes in this study is indicated throughout this review. Additionally, the authors of this study did not specify the scoring for any components in the MDS-2003, but did specify the scoring of components in the PREDIMED score and MDS-2006 [23].

#### 3.4.1. Vegetables

All studies included vegetables as a beneficial component in their indexes [20,21,22,23,24,25,26,27,28,29,30,31,32,33,34,35,36,37,38,39,40,41,42,43,44,45,46]. Only four studies discriminated between vegetables and potatoes establishing both as separate categories in their indexes [23,25,26,38]. In all four studies, vegetables and potatoes were both considered beneficial components [23,25,26,38]. Two of these studies employed the MDS-2006 [23,38], which by design includes potatoes as a separate component from vegetables [14]. Castro-Rodriguez et al. (2010) and (2016) categorized potatoes as a separate category in their modified version of the MDS-2004 [25,26]. In regard to the studies not including a separate category for potatoes [20,21,22,23,24,27,28,29,30,31,32,33,34,35,36,37,39,40,41,42,43,44,45,46], only one study explicitly reported the inclusion of root vegetables in the vegetable component of their index [24], which was not clear in the other studies [20,21,22,23,27,28,29,30,31,32,33,34,35,36,37,39,40,41,42,43,44,45,46]. Interestingly, the two studies utilizing Khoury’s criteria grouped vegetables and fruits together into a single category [31,35].

#### 3.4.2. Fruits

All studies included fruits as a beneficial component in their index [20,21,22,23,24,25,26,27,28,29,30,31,32,33,34,35,36,37,38,39,40,41,42,43,44,45,46]. Seven studies specified the inclusion of fruit juices in the fruit component [20,33,40,41,42,44,46]. Only one study explicitly excluded fruit juices from inclusion in the fruit component of their index [22]. Moreover, nine studies grouped fruits and nuts together as a single component [21,22,24,27,28,29,30,37,39] in line with the original designs of the MDS-2003 [13] and rMed [49] employed by these studies.

#### 3.4.3. Nuts

22 studies included nuts as a beneficial component in their indexes [20,21,22,23,24,27,28,29,30,32,33,34,36,37,39,40,41,42,43,44,45,46], including only the PREDIMED score of Martínez-Galiano et al. (2018) [23]. Thirteen studies categorized nuts as a separate component from fruits [20,23,32,33,34,36,40,41,42,43,44,45,46]. In regard to the studies excluding nuts [23,25,26,31,35,38], two used the MDS-2006 [23,38] and two Khoury’s criteria [31,35], both of which omit nuts from their indexes [14,51]. Castro-Rodriguez et al. (2010) and (2016) excluded nuts from their modified version of the MDS-2004 [25,26]. Although the MSD-2003 traditionally includes nuts with fruit in its initial categorization [13], Martínez-Galiano et al. (2018) only listed fruits, without nuts, in their computation of the MDS-2003 [23].

#### 3.4.4. Cereals

21 studies included cereals a beneficial component in their index [20,21,22,23,24,25,26,27,28,29,30,32,33,34,36,37,38,39,42,45,46], including only the MDS-2003 and MDS-2006 of Martínez-Galiano et al. (2018) [23]. The studies excluding cereals utilized the PREDIMED score [23,40,41,43,44] and Khoury’s criteria [31,35], both of which omit cereals from their indexes [15,51]. Eight studies incorporated specifically non-refined or whole grains into their indexes, rather than cereals as one all-encompassing component [20,23,32,33,34,36,38,45]. Two of these studies employed the MDS-2006 [23,38], which traditionally incorporates only non-refined grains into its index [14]. The remaining six studies utilized the MDS-2003 [20,32,33,34,36,45], which does not discriminate between refined and non-refined grains in its initial categorization of cereals [13]. Only one study clearly confirmed the incorporation of both refined and whole grain flours in their cereal component [22]. Moreover, four studies additionally included pasta and rice as separate components from cereals in their indexes [25,26,42,46]. Castro-Rodriguez et al. (2010) and (2016) included pasta and rice as separate components [25,26], whereas the two studies using the modified KIDMED index combined pasta and rice together into a single category [42,46]. Regardless, pasta and rice components were also scored as beneficial components [25,26,42,46].

#### 3.4.5. Legumes

25 studies included legumes as a beneficial component in their indexes [20,21,22,23,24,25,26,27,28,29,30,32,33,34,36,37,38,39,40,41,42,43,44,45,46]. Only five studies provided a description of the legume component [20,23,24,33,39]. One study listed “baked beans, pulses, bean curd, tahini, and soya meat” [24]. Two studies reported using the same computation of the MDS-2003 and as such included “dried beans, lentils, peas, soups (i.e., split pea), tofu, and soymilk” [20,33]. Another study detailed kidney beans, green peas, and lentils as being included in the component [39]. Martínez-Galiano et al. (2018) provided brief descriptions for legumes in the PREDIMED score (peas, lentils, and beans) and MDS-2006 (e.g., peas and beans) but noticeably omitted a description of legumes in the MDS-2003 [23]. Of the two studies excluding legumes [31,35], legumes were excluded in accordance with Khoury’s criteria [51].

#### 3.4.6. Fish

All studies included fish as a beneficial component in their indexes [20,21,22,23,24,25,26,27,28,29,30,31,32,33,34,35,36,37,38,39,40,41,42,43,44,45,46]. 10 studies expanded the fish component to additionally encompass either shellfish [20,24,33,40,41] or seafood [22,28,29,43,44].

#### 3.4.7. Meat

25 studies included meat in their indexes [20,21,22,23,24,25,26,27,28,29,30,31,32,33,34,35,36,37,38,39,40,41,43,44,45]. Two studies using the MDS-2006 discriminated between meat and poultry establishing both as separate categories [23,38]. Conversely, one study using Khoury’s criteria reported excluding poultry from the meat component of their index [35]. Moreover, five studies limited the meat component to entail only red and processed meat [20,32,33,34,36], thereby, deviating from the initial categorizations of the MDS-2003 and aMed [13,59].

Almost all studies with a meat component scored meat, poultry, or red and processed meat as detrimental components [20,21,22,23,24,25,26,27,28,29,30,31,32,33,34,35,36,37,38,39,45], including only the MDS-2003 and MDS-2006 of Martínez-Galiano et al. (2018) [23]. However, the five studies utilizing the PREDIMED score scored the consumption of chicken, turkey, or rabbit instead of veal or beef, pork, hamburgers, or sausages as contributing to increased adherence to the MD [23,40,41,43,44]. Only in a separate question of the PREDIMED score was the consumption of >1 serving of red meat, hamburgers, or other meat products (e.g., sausages) scored as contributing to decreased adherence to the MD [23,40,41,43,44]. Importantly, the two studies excluding meat from their index utilized the modified KIDMED index [42,46], which does not include meat in its index [55].

#### 3.4.8. Dairy Products

24 studies included dairy products as a component in their indexes [20,21,22,23,24,25,26,27,28,29,30,32,33,34,37,38,39,40,41,42,43,44,45,46]. Two studies using the MDS-2006 discriminated among dairy products to only include full-fat dairy [38] or dairy with fat [23]. The five studies employing the PREDIMED score did not include a general dairy component but instead butter and cream grouped together with the non-dairy food of margarine in their index [23,40,41,43,44]. Similarly, Castro-Rodriguez et al. (2010) and (2016) listed only milk in their index [25,26].

Overall, the scoring of dairy products was almost evenly split. 13 studies scored dairy products as a detrimental component [21,22,23,25,26,30,38,39,40,41,43,44,45], which remains in alignment with the original designs of their respective indexes [13,14,15,47,49]. Contrastingly, 11 studies designated dairy products a beneficial component [20,24,27,28,29,32,33,34,37,42,46]. Of these 11 studies, nine utilized the MDS-2003 [20,24,27,28,29,32,33,34,37] and as such modified the original scoring system to score dairy products as a beneficial component given nutritional requirements during pregnancy [13]. The other two remaining studies employed the modified KIDMED index [42,46], which by design scores dairy products as a beneficial component [55]. The three studies excluding dairy products utilized Khoury’s criteria [31,35] and aMed [36], therefore, excluding dairy products in accordance with these indexes [51,59].

#### 3.4.9. Alcohol

Since alcohol is not generally recommended during pregnancy, 19 studies subsequently removed alcohol from their indexes [20,21,23,24,25,26,27,28,29,30,32,33,34,36,39,40,41,43,45] and four studies [31,35,42,46] excluded alcohol in accordance with the original design of their indexes [51,55]. Of the four studies maintaining alcohol in their indexes [22,37,38,44], three studies regarded alcohol as a detrimental component [22,37,38], thereby, modifying the scoring of their original indexes [13,14,49]. Specifically, Saunders et al. (2014) modified the MDS-2003 and scored less alcohol consumption as contributing to increased adherence to the MD [37]. Babili et al. (2020) utilized the MDS-2006 and scored less (<100 mL), but importantly not no, alcohol consumption as contributing highest to increased adherence to the MD [38]. Contrastingly, Jardí et al. (2019) used rMed and scored alcohol consumption dichotomously with no alcohol consumption contributing to increased adherence to the MD in comparison to any alcohol consumption [22]. Importantly, Tomaino et al. (2020) was the only study to maintain that greater wine intake contributes to increased adherence to the MD in their index [44] as designated by the PREDIMED score [15].

#### 3.4.10. Lipid Ratios

Beyond food components, 10 studies incorporated lipid ratios as a component in their indexes [20,23,28,29,32,33,34,36,37,45], including only the MDS-2003 of Martínez-Galiano et al. (2018) [23]. Eight studies included a ratio of MUFA:SFA [20,23,28,29,33,36,37,45] in accordance with the designs of the MDS-2003 and aMED [13,59]. Going further, the other two studies included a ratio of unsaturated fatty acids (USFA):SFA [32,34]. Interestingly, both of these studies also employed the MDS-2003 [32,34] but subsequently modified the index to replace the ratio of MUFA:SFA with a ratio of USFA:SFA [13]. In most cases, a ratio of MUFA:SFA or USFA:SFA was considered a beneficial component [20,32,33,34,36,45]. Nevertheless, four studies did not explicitly report whether the lipid ratio was considered as a beneficial or detrimental component in their indexes [23,28,29,37].

Of the studies excluding a lipid ratio [21,22,23,24,25,26,27,30,31,35,38,39,40,41,42,43,44,46], 13 studies incorporated olive oil into their respective indexes [21,22,23,30,31,35,38,40,41,42,43,44,46], including only the PREDIMED score and MDS-2006 of Martínez-Galiano et al. (2018) [23]. Nevertheless, three studies utilized the MDS-2003 [24,27,39] and two a modified version of the MDS-2004 [25,26], thereby, excluding an original component from their indexes [13,47].

#### 3.4.11. Other Food Components

Besides the aforementioned food components traditional to the original designs of the MD [12,13], several studies also included additional food components in their indexes.

##### Olive Oil and Rapeseed Oil

13 studies included olive oil as a beneficial component as designated in their respective indexes [21,22,23,30,31,35,38,40,41,42,43,44,46], including the MDS-2006 [14], rMed [49], PREDIMED score [15], modified KIDMED index [55], and Khoury’s criteria [51]. Of the two studies utilizing Khoury’s criteria, olive oil was grouped with rapeseed oil, which together were considered a beneficial component [31,35]. Importantly, none of the studies including olive or rapeseed oil in their index additionally included a lipid ratio [21,22,23,30,31,35,38,40,41,42,43,44,46].

##### Fast Food and Junk Food

Five studies included fast food as a detrimental component in their indexes [25,26,39,42,46]. Castro-Rodriguez et al. (2010) and (2016) added a generic fast food component to their modified version of the MDS-2004 [25,26] encompassing “candies, industry pastry, precooked pizzas, fried food”, and fast food hamburgers [25,26]. Similarly, de Batlle et al. (2008) added a junk food and fat component to the MDS-2003, which collectively grouped fast food hamburgers with desserts (i.e., pastries, candies, and ice cream), snacks (i.e., chips and popcorn), as well as butter and margarine [39]. Contrastingly, the two studies employing the modified KIDMED index incorporated the food habit component of visiting a fast food (“hamburger”) restaurant each week [42,46].

##### Sweets, Candies, and Pastries

Seven studies included sweets, candies, and pastries as detrimental components in their indexes [23,40,41,42,43,44,46]. Notably, the two studies utilizing the modified KIDMED index included two separate categories: One for commercial baked goods and pastries and the other for sweets and candy [42,46]. Interestingly, both categories were completely divorced from the fast food component [42,46], unlike the three studies previously discussed which included sweets, pastries, and desserts in the fast food component [25,26,39]. The five studies using the PREDIMED score incorporated a component for commercial pastries, which did not include sweets or candies [23,40,41,43,44].

##### Sofrito

The five studies utilizing the PREDIMED score [23,40,41,43,44] included sofrito in their indexes [15]. Sofrito was described as a sauce consisting of tomato, garlic, onion, and leeks or peppers sautéed in olive oil [23,40,44]. If dishes were consumed with sofrito, it was considered as a beneficial component [23,40,41,43,44].

##### Sweetened or Carbonated Beverages or Coffee

Five studies included sweetened or carbonated beverages [23,40,41,43,44] and two studies included coffee [31,35] in alignment with their respective indexes [15,51]. The five studies including sweetened, carbonated beverages in their indexes all employed the PREDIMED score and designated this component as detrimental [23,40,41,43,44]. The two studies including coffee used Khoury’s criteria and similarly scored drinking more than two servings of coffee a day as a detrimental component [31,35].

##### Skipping Breakfast

The two studies using the modified KIDMED index included a food habit component on skipping breakfast as a detrimental component in their index [42,46].

### 3.5. Scoring Systems

#### 3.5.1. Cut-Off Points

In terms of scoring, approximately half of the studies used cut-off points based on the distribution of intake in the study population [21,22,23,24,27,28,29,30,32,34,36,37,39,45], whereas the other studies utilized fixed criteria, including predetermined food servings and food habits, as cut-off points for food and lipid components in their indexes [20,23,25,26,31,33,35,38,40,41,42,43,44,46]. Both Table 2 and Table 3 provide details regarding the cut-off points specific to the included studies.

##### By Distribution

In total, 14 studies used cut-off points based on the distribution of intake in the study population [21,22,23,24,27,28,29,30,32,34,36,37,39,45], including only the MDS-2003 of Martínez-Galiano et al. (2018) [23]. 11 studies reported using the median intake values of the study population for components [23,24,27,28,29,32,34,36,37,39,45], therefore, remaining in alignment with the cut-off points designated in the MDS-2003 or aMed [13,59]. In both indexes, women with higher intakes of beneficial foods receive +1 and women with lower intakes of beneficial foods receive 0 with the opposite scoring being applied to detrimental foods [13,59]. Conversely, the three remaining studies employed rMed [21,22,30], thereby, using tertiles of intake based on the study population as cut-off points [49]. In all three studies, each component was expressed in energy density (1000 kcal/day), partitioned into tertiles, and then assigned a value of 0, +1, or +2 in which higher intakes of beneficial foods received higher scoring and higher intakes of detrimental foods received lower scoring [21,22,30].

##### By Fixed Criteria

Alternatively, 14 studies used fixed criteria as cut-off points for components [20,23,25,26,31,33,35,38,40,41,42,43,44,46], including only the PREDIMED score and MDS-2006 of Martínez-Galiano et al. (2018) [23]. The nine studies employing the PREDIMED score [23,40,41,43,44], Khoury’s criteria [31,35], and modified KIDMED index [42,46] used a combination of fixed servings of components (e.g., ≥2 servings vegetables per day) as well as food habits (e.g., use of olive oil as principle fat) requiring a “yes/no” answer in their indexes [15,51,55]. In the PREDIMED score and Khoury’s criteria, women receive +1 for meeting the fixed criteria of beneficial components and 0 for exceeding the criteria for detrimental components [15,51]. In the KIDMED index, women receive +1 for meeting the fixed criteria for beneficial components but −1, rather than 0, for exceeding the fixed criteria of detrimental components [55]. In contrast, the two studies using the MDS-2006 scored components 0, +1, +2, +3, +4, or +5 using only fixed servings per month in which higher servings of beneficial components received higher scoring and higher intakes of detrimental components received lower scoring [23,38].

Interestingly, four studies deviated from their respective indexes to utilize fixed consumption of components as cut-off points [20,25,26,33]. In their modified version of the MDS-2004, Castro-Rodriguez et al. (2010) and (2016) utilized fixed servings and scored components 0, +1, or +2 with higher intakes of beneficial foods receiving higher scoring and higher intakes of detrimental foods receiving lower scoring [25,26]. Rather than using median intake values characteristic to the MDS-2003 [13], Chatzi et al. (2017) and Lindsay et al. (2020) applied fixed servings sizes based on pregnancy recommendations in the 2010 Dietary Guidelines for Americans to each component in their index [20,33].

#### 3.5.2. Range of Scores

Although all studies reported a higher final score indicating higher adherence to MD and a lower score indicating less adherence, the range of scores varied greatly depending upon the chosen index and subsequent modifications to components [20,21,22,23,24,25,26,27,28,29,30,31,32,33,34,35,36,37,38,39,40,41,42,43,44,45,46]. Overall, the range of scores ranged from 0–5 [31,35] to 0–55 [38]. A total of eight studies maintained the original range of scores designated in their respective indexes [22,31,35,37,38,42,44,46] of which five also maintained the original scoring of all components [31,35,42,44,46] and three modified alcohol to be scored as a detrimental component in their indexes [22,37,38].

Despite alcohol being a traditional component in their indexes [13,14,15,47,49,59], 19 studies removed alcohol from their indexes [20,21,23,24,25,26,27,28,29,30,32,33,34,36,39,40,41,43,45]. Subsequently, five studies separated fruits and nuts into two distinct categories [20,32,33,34,45], and three studies excluded MUFA:SFA from their indexes [24,27,39]. Only three studies added completely new components to their index, including potatoes, pasta, rice, and fast food [25,26] and junk food and fat [39].

## 4. Discussion

In summary, the findings of this narrative review underscore the disparity present in the definitions of the MD operationalized in *a priori* indexes to assess maternal adherence to the MD during pregnancy. Among the 27 studies [20,21,22,23,24,25,26,27,28,29,30,31,32,33,34,35,36,37,38,39,40,41,42,43,44,45,46], eight different *a priori* indexes were identified of which the most commonly used index was the MDS-2003 (*n* = 12) [20,23,24,27,28,29,32,33,34,37,39,45]. Studies included a range of 5 [31,35] to 13 [42,44,46] dietary components in their indexes; however, a total of 17 different dietary components were identified among the indexes of the 27 studies. 10 components were considered traditional to the original indexes of the MD [12,13]. Only three components—vegetables, fruits, and fish—were common among all indexes and scored similarly as beneficial components. Approximately half of the studies utilized cut-off points based on the distribution of intake levels in the study population [21,22,23,24,27,28,29,30,32,34,36,37,39,45]; the other studies utilized fixed criteria [20,23,25,26,31,33,35,38,40,41,42,43,44,46]. The range of scores varied from 0–5 [31,35] to 0–55 [38] in which zero indicated the lowest adherence to the MD and the highest number indicated the highest adherence.

Beyond the number of components, the degree of discrimination among food components was a major source of heterogeneity. First, nine studies grouped fruits and nuts together into a single category [21,22,24,27,28,29,30,37,39] in line with the initial categorization of their indexes [13,49]. Although minimally processed fruits and nuts are both constituents in many healthful dietary patterns [60], nutritionists and dietitians generally regard fruits and nuts as two distinct foods composed of different nutritional properties as evidenced by separate categorizations and recommendations in countries’ dietary guidelines [17,61]. While separating fruits and nuts is therefore in agreement with such dietary guidelines, it could affect the scoring system by perhaps disproportionately weighing the importance of these foods over the other dietary components in the indexes. Second, four studies separated vegetables and potatoes into separate components in their indexes [23,25,26,38]. Only one study explicitly reported the inclusion of root vegetables in the vegetable component [24], whereas for the other studies this was not clearly stated [20,21,22,23,27,28,29,30,31,32,33,34,35,36,37,39,40,41,42,43,44,45,46]. Although some countries such as the United States group vegetables and potatoes together in their dietary guidelines [17], a distinction between these foods in *a priori* indexes is likely relevant for pregnancy given results showing increased risk of GDM in women with higher pre-pregnancy consumption of potatoes [62]. Third, eight studies distinguished among the cereal component to include only non-refined or whole grains as a beneficial component in their indexes [20,23,32,33,34,36,38,45]. In general adult populations, it was shown that consumption of non-refined, rather than refined, grains was associated with reduced risk of coronary heart disease [63] and type 2 diabetes mellitus [64]. Furthermore, recent research has shown that consumption of refined grains during pregnancy by women with GDM was associated with risk of increased body mass index z-score and overweight and obesity in offspring at 7 years (yr) [65]. This complicates the precedent of utilizing one all-encompassing component for cereals in many *a priori* indexes [12,13]. Additionally, five studies included red and processed meat as a detrimental component in their index as opposed to one all-encompassing component for meat [20,32,33,34,36]. In general adult populations, red and processed meat consumption increases the risk of CVD and all-cause mortality [66,67], particularly in comparison to white-meat (e.g., poultry) [60,66]. Although comparatively research during pregnancy is limited, research has shown the higher pre-pregnancy consumption of a Western dietary pattern, explained largely by red and processed meat products, was associated with increased risk of GDM in comparison to a more prudent dietary pattern, featuring high intakes of vegetables and fruit as well as poultry and fish [68].

Aside from discrimination among components, considerable discrepancies also arose from amendments to dairy and alcohol—the only components to be modified in light of pregnancy. According to the initial scoring of most indexes included in this review, dairy stands as a detrimental component contributing to decreased adherence to the MD [13,14,15,47,49]. This likely stems from the low to moderate consumption of dairy products observed in the traditional MD [12,13] as well as the controversy surrounding reducing SFA intake for CVD prevention [69]. Nevertheless, nine studies deviated from this precedent to score dairy as a beneficial component [20,24,27,28,29,32,33,34,37], likely due to increased calcium requirements during pregnancy [19]. Although no clear consensus has been reached, recent research has also reported a neutral association between total dairy consumption and CVD and an inverse association with type 2 diabetes mellitus [70,71]. This provides a further reason that the scoring of dairy products should be reconsidered in pregnant and non-pregnant populations alike. Moreover, given the widespread recommendation to abstain from alcohol during pregnancy [17,18], it remains unsurprising that 19 studies deviated from their indexes to exclude alcohol [20,21,23,24,25,26,27,28,29,30,32,33,34,36,39,40,41,43,45]. Nevertheless, future studies should be cautious about reflexively excluding alcohol from their index as this could misclassify some pregnant participants who do consume alcohol during pregnancy as non-consumers. If alcohol consumption is reported by pregnant participants in a cohort, reversing the scoring of alcohol to be considered a detrimental component could also serve as a viable option as performed in three studies in this review [22,37,38]. Although even in this approach, future studies should take into account that then very low alcohol consumption could inordinately influence the scoring system as much as low intakes of vegetables or high intakes of red and processed meat given these indexes allot equal weight to each individual dietary component.

Beyond the components themselves, differences in cut-off points further challenged the comparability of indexes. Although the original designs of the MD utilize the median intake values of the study population as cut-off points [12,13], the use of median intake levels ignores the extremes of intake within a population [72]. As a result, it ignores variance of intake within a population and erroneously assumes homogeneity of risk among all intake levels [73]. Furthermore, establishing cut-off points based on the distribution of intake within a study population limits comparability between studies, which proves particularly cumbersome for comparisons between Mediterranean and non-Mediterranean countries due to vastly different dietary habits [74]. While fixed criteria could serve as a possible solution to comparability among studies, it should be noted that the use of fixed criteria is limited in its ability to distinguish among intake levels when components are rarely consumed in a population (e.g., alcohol consumption during pregnancy). Furthermore, it is important to note that FFQs are a common dietary assessment method used to measure dietary intake in large epidemiological studies [74]. Instead of capturing absolute intake, FFQs operate by accurately ranking individuals based on habitual intake levels [75]. Therefore, basing cut-off points on the distribution remains consistent with the use of FFQs [74]. If tertiles of intake are utilized as employed in rMed, it would allow for the additional advantage of better assessing the variance of intake within a study population by resulting in a wider range of scores than possible using median intake levels alone [49].

In regard to dietary assessment methods, variability was evident in both the choice and the time period in which dietary data was collected. Most studies employed an FFQ [20,21,22,23,24,25,26,27,28,29,30,31,32,34,35,36,37,38,39,40,41,43]; two studies utilized repeated 24-h recalls [33,45]; and three used index-specific questionnaires [42,44,46]. Given that each of these methods are self-reported, information bias (e.g., recall bias) remains an ever-present possibility obscuring the accurate assessment of dietary intake by participants [76]. However, even more so, only two studies measured dietary intake during each trimester of pregnancy [22,33]. While it has been reported that dietary intake patterns do not drastically change during pregnancy [77], physiological changes during pregnancy (e.g., nausea) may alter normal food consumption over short periods of time [78], which may not be as readily captured within the timeframe of an FFQ designed to capture habitual food intake [75]. Therefore, if feasible within study constraints, studies should consider utilizing repeated 24-h recalls, an FFQ multiple times during each trimester, or a combination of the two in order to better ensure a truly accurate estimation of dietary intake during pregnancy.

Lastly, it is worth emphasizing that only one index in this review—Khoury’s criteria—was specifically developed for use during pregnancy [51]. Conversely, most of the other indexes were validated in the general adult population [13,14,49,59]. Specifically, the PREDIMED score was developed for use in older adults (55–80 years) at high-risk for CVD [15]. Alternatively, the KIDMED index was validated in younger individuals (2–24 years), therefore, explaining the exclusion of alcohol from its index [55]. Besides the incongruity of some traditional components of the MD to pregnancy (e.g., alcohol), adherence to the MD does not necessitate many important nutritional recommendations during pregnancy essential to fetal growth and development, including adequate micronutrient intake and sufficient hydration. Importantly, inadequate intake of micronutrients during pregnancy results in adverse offspring health outcomes, including neural tube defects from inadequate intake of folate [79]. To date, however, only the MDS for pregnancy has incorporated folic acid, iron, and calcium into their *a priori* index; yet, this index and these micronutrient considerations were not represented in any of the studies in this review [16]. Similarly, insufficient hydration during pregnancy has also been shown to be associated with adverse offspring health outcomes ranging from neural tube defects to musculoskeletal and congenital heart defects [80]. While untraditional to the original indexes of the MD [12,13], the aforementioned nutritional recommendations during pregnancy should be considered for incorporation into *a priori* indexes assessing maternal adherence to the MD during pregnancy when studying the MD as a proxy for a healthy diet during pregnancy due to their immense importance in fetal growth and development [79,80].

However, ultimately it must be emphasized that MD is a descriptive diet reflecting the common dietary characteristics traditional to populations in the Mediterranean basin [1,2]. As such, it is not a standard diet quality index designed to ensure optimal health based on the most current nutrition knowledge [1,81]. Similar to arguments concerning the incorporation of modern food components (e.g., fast food and desserts) [39,55], the addition of components specific to nutritional requirements during pregnancy would amplify the relevancy of *a priori* indexes assessing adherence to the MD in today’s society. Yet, these additions would consequently transition these *a priori* indexes away from actually measuring the MD. Therefore, future studies must carefully weigh these advantages and disadvantages of modifying *a priori* indexes for use in pregnant populations evaluating whether they are truly interested in measuring adherence to the MD or simply adherence to a healthy diet quality index for pregnancy.

## 5. Conclusions

In conclusion, this narrative review emphasizes the vast heterogeneity and subjectivity present among the components, cut-off points, and scoring systems of *a priori* indexes used to assess maternal adherence to the MD during pregnancy. As a dietary pattern, the MD maintains the key advantage of encapsulating the synergistic effects of nutrients, which would otherwise remain undetectable when investigating single nutrients alone [9,60]. Although *a priori* indexes provide a useful means to assess maternal adherence to the MD during pregnancy, future studies should carefully examine the importance of addressing pregnancy-specific nutritional recommendations in *a priori* indexes, including altering the scoring of dairy and alcohol and incorporating micronutrients (e.g., folate) and water. While such modifications may be untraditional to the original indexes of the MDS [12,13], it would allow studies investigating the MD as a proxy for a healthy diet in pregnancy a better approach to address pregnancy-specific nutritional requirements crucial for optimal maternal and offspring health.

## Figures and Tables

**Figure 1 nutrients-13-00582-f001:**
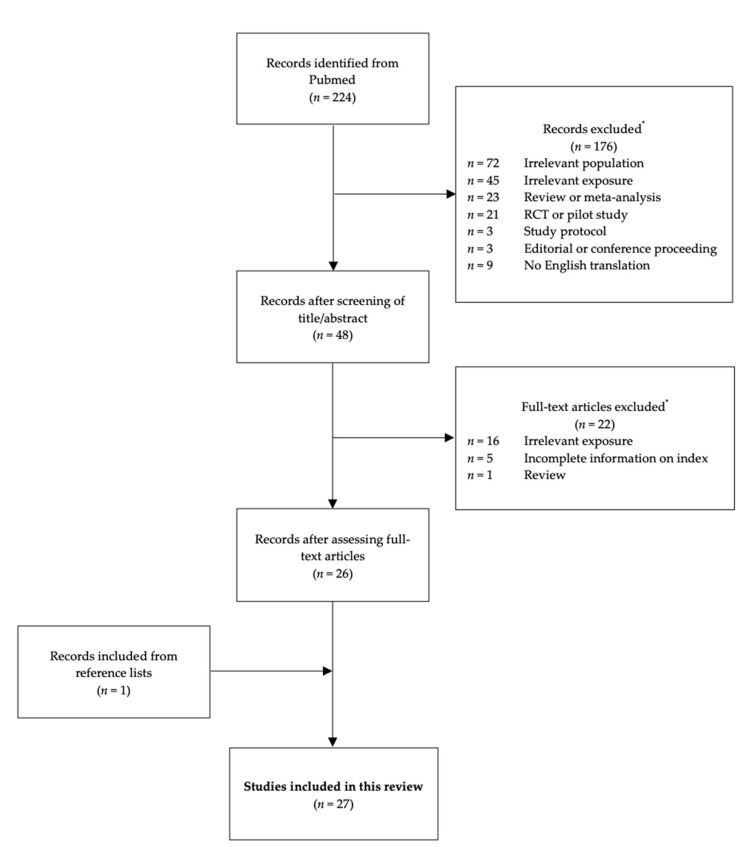
Flowchart of identification, screening, and inclusion of studies using *a priori* indexes to assess maternal adherence to the MD during pregnancy. * Exclusion criteria for review: (1) Irrelevant population (e.g., non-pregnant, pre-pregnant, or peri-conceptional study population), (2) irrelevant exposure (i.e., exposure not pertaining to adherence to MD), (3) incomplete information on the *a priori* index in methods, (4) non-observational study design study (e.g., RCT), (5) reviews, meta-analyses, editorials, or conference proceedings, and (6) no English translation available. Abbreviation: MD, Mediterranean diet; RCT, randomized controlled trial.

**Table 1 nutrients-13-00582-t001:** Search strategy.

Search Number	Pubmed: 1 July 2020	Search Results
1	mother OR maternal OR pregnan * OR gestation * OR prenatal	1,309,142
2	mother [MESH]	43,260
3	maternal nutritional physiological phenomena [MESH]	5891
4	pregnancy [MESH]	890,464
5	Search 1 OR 2 OR 3 OR 4	1,320,879
6	“diet, Mediterranean” OR “diets, Mediterranean” OR “Mediterranean diet” OR “Mediterranean diets” OR “Mediterranean diet score” OR “Mediterranean diet scores” OR “Mediterranean dietary pattern” OR “Mediterranean dietary patterns”	5706
7	“diet, Mediterranean” [MESH]	3371
8	Search 6 OR 7	5706
9	Search 5 AND 8	224

* indicates the truncation of a term used to broaden search.

**Table 2 nutrients-13-00582-t002:** Overview of twenty-seven studies using an *a priori* index to assess maternal adherence to the MD during pregnancy.

Author, Year	Study Design, Location	Study Population	Dietary Assessment	*A priori* Index	Food and Nutrient Components *	Scoring	Range ^†^	Main Outcomes
Babili et al. (2020) [38]	Cross-sectional study, Greece	535 mother-child pairs	Semi-quantitative, 69-question FFQ, post-partum	MDS-2006 [14]	11 components: vegetables (+) potatoes (+) fruits (+) non-refined cereals (+) legumes (+) fish (+) meat (−) poultry (−) full-fat dairy (−) alcohol (−) olive oil (+)	Fixed criteria (0–5) ^§^	0–55	No association shown between MDS and infant weight class or length class at birth.
Bédard et al. (2020) [24]	Population-based cohort (ALSPAC), UK	8907 mother-child pairs	43-item FFQ, T3	MDS-2003 [27]	7 components vegetables (+) fruits and nuts (+) cereals (+) legumes (+) fish and shellfish (+)meat (−) dairy (+)	Median intake (0,1)	0–7	Positive association shown between MDS and FEF_25–75%_ z-scores in offspring at 8.5 years.
Castro-Rodriguez et al. (2010) [25]	Longitudinal cohort (EISL), Spain	1409 children **	Semi-quantitative, 11-item FFQ, post-partum	Modified MDS-2004 [47]	11 components: vegetables (+) potatoes (+) fruits (+) cereals (+) pasta (+) rice (+) legumes (+) fish (+) meat (−) milk (−) fast foods (−)	Fixed criteria (0–2) ^§^	0–22	No association shown between MDS and wheezing in offspring 12 months after birth.
Castro-Rodriguez et al. (2016) [26]	Longitudinal cohort (EISL), Spain	1000 children **	Semi-quantitative, 11-item FFQ, post-partum	Modified MDS-2004 [47]	11 components: vegetables (+) potatoes (+) fruits (+) cereals (+) pasta (+) rice (+) legumes (+) fish (+) meat (−) milk (−) fast foods (−)	Fixed criteria (0–2) ^§^	0–22	No association shown between MDS and current wheeze, rhinitis, or eczema 12 months after birth.
Chatzi et al. (2008) [27]	Cohort study, Spain	460 mother-child pairs	Semi-quantitative, 42-item FFQ ^††^	MDS-2003 [13]	7 components: vegetables (+) fruits and nuts (+) cereals (+) legumes (+) fish (+) meat (−) dairy (+)	Median intake (0,1) ^§§^	0–7	Inverse association shown between MDS and risk of persistent wheeze, atopic wheeze, and atopy in offspring at 6.5 year.
Chatzi et al. (2012) [28]	Population-based cohort studies (INMA), Spain and (RHEA), Greece	2461 mother-child pairs (INMA); 889 mother-child pairs (RHEA)	Semi-quantitative, 100-item FFQ, T1 (INMA); semi-quantitative, 250-item FFQ, T2 (RHEA)	MDS-2003 [13]	8 components: vegetables (+) fruits and nuts (+) cereals (+) legumes (+) fish and seafood (+) meat (−) dairy (+) MUFA:SFA (.)	Median intake (0,1) ^§§^	0–8	Inverse association shown between MDS and risk of FGR for weight in infants at birth in the INMA-Mediterranean cohort.
Chatzi et al. (2013) [29]	Population-based cohort studies (INMA), Spain and (RHEA), Greece	1771 mother-child pairs (INMA); 745 mother-child pairs (RHEA)	Semi-quantitative, 100-item FFQ, T1 (INMA); semi-quantitative, 250-item FFQ, T2 (RHEA)	MDS-2003 [13]	8 components: vegetables (+) fruits and nuts (+) cereals (+) legumes (+) fish and seafood (+) meat (−) dairy (+) MUFA:SFA (.)	Median intake (0,1) ^§§^	0–8	No association shown between MDS and wheeze and eczema 12 months after birth.
Chatzi et al. (2017) [20]	Prospective mother-child cohort study (Viva), USA; population-based cohort study (RHEA), Greece	997 mother-child pairs (Viva); 569 mother-child pairs (RHEA)	Semi-quantitative, 146-item FFQ, T1 (Viva); semi-quantitative, 250-item FFQ, T2 (RHEA)	MDS-2003 [13]	9 components: vegetables (+) fruits (+) nuts (+) whole grains (+) legumes (+) fish and shellfish (+) red and processed meat (−) dairy (+) MUFA:SFA (+)	Fixed criteria (0,1) ^§^	0–9	Inverse association between MDS and BMI z-score, waist circumference, skin-fold thickness, SBP, and DBP in offspring.
de Batlle et al. (2008) [39]	Cross-sectional study (ISSAC), Mexico	1476 children **	Semi-quantitative, 70-item FFQ, post-partum	MDS-2003 [13]	8 components: vegetables (+) fruits and nuts (+) cereals (+) legumes (+) fish (+) meat (−) dairy (−) junk food and fat (−)	Median intake (0,1)^¶¶^	0–8	Inverse association between MDS and current sneezing in offspring 6–7 year.
Fernández-Barrés et al. (2016) [21]	Population-based cohort study (INMA), Spain	1827 mother-child pairs	Semi-quantitative, 101-item FFQ, T1 and T3	rMed [48]; aMed ***	8 components: vegetables (+) fruits and nuts (+) cereals (+) legumes (+) fish (+) meat (−) dairy (−) olive oil (+)	Tertiles of intake (0–2)	0–16	Inverse association shown between MDS and waist circumference in offspring at 4 year.
Fernández-Barrés et al. (2019) [30]	Population-based cohort study (INMA), Spain	2195 mother-child pairs	Semi-quantitative, 101-item FFQ, T1 and T3	rMed [49]	8 components: vegetables (+) fruits and nuts (+) cereals (+) legumes (+) fish (+) meat (−) dairy (−) olive oil (+)	Tertiles of intake (0–2)	0–16	Inverse association between MDS and risk of having an offspring of larger size followed by an accelerated gain in BMI.
Gesteiro et al. (2012) [40]	Cross-sectional study, Spain	35 mother-child pairs	169-item FFQ, T2 and post-partum	PREDIMED score [50]	12 components:^†††^ vegetables (+) fruits (+) nuts (+) legumes (+) fish and shellfish (+) chicken, turkey, or rabbit (+) red meat, hamburger, or other meat products (−) butter, margarine, or cream (−) olive oil (+) commercial sweets or pastries (−) dishes with sofrito (+) ^§§§^ sweet or carbonated beverages (−)	Fixed criteria (0,1) ^¶¶¶^	0–13	Inverse association shown between MDS and core-blood glycemia and insulinemia in infants.
Gesteiro et al. (2015) [41]	Cross-sectional study, Spain	35 mother-child pairs	169-item FFQ, T2 and post-partum	PREDIMED score [50]	12 components: ^†††^ vegetables (+) fruits (+) nuts (+) legumes (+) fish and shellfish (+) chicken, turkey, or rabbit (+) red meat, hamburger, or other meat products (−) butter, margarine, or cream (−) olive oil (+) commercial sweets or pastries (−) dishes with sofrito (+) ^§§§^ sweet or carbonated beverages (−)	Fixed criteria (0,1) ^¶¶¶^	0–13	Inverse association shown between MDS and cord-blood LDL-c, Apo B, homocysteine, and Apo A1/Apo B ratio in infants.
Haugen et al. (2008) [31]	Population-based cohort study (MoBa), Norway	26,563 mother-child pairs	Semi-quantitative, 255-item FFQ, T2	Khoury’s criteria [51]	5 components: vegetables and fruits (+) fish (+) meat (−) olive oil or rapeseed oil (+) coffee (−)	Fixed criteria (0,1) ^¶¶¶^	0–5	No association shown between MDS and PTB in infants at birth.
Izadi et al. (2016) [45]	Case-control hospital-based study, Iran	459 mothers	Repeated 24-hour recalls ^††^	MDS-2003 [13]	9 components: vegetables (+) fruits (+) nuts (+) whole grains (+) legumes (+) fish (+) meat (−) dairy (−) MUFA:SFA (+)	Median intake (0,1) ^§§^	0–9	Inverse association shown between MDS and GDM.
Jardí et al. (2019) [22]	Cohort study (ECLIPSES study), Spain	513 mothers	45-item FFQ, T1, T2, and T3	rMed [21]	9 components: vegetables (+) fruits and nuts (+) cereals (+) legumes (+) fish and seafood (+) meat (−) dairy (−) alcohol (−) olive oil (+)	Tertiles of intake (0–2) ****	0–18	No association shown in MDS across trimesters of pregnancy.
Lange et al. (2010) [32]	Cohort study (Viva), USA	1376 mother-child pairs	Semi-quantitative, 166-item FFQ, T1 and T2	MDS-2003 [13]	9 components: vegetables (+) fruit (+) nuts (+) whole grains (+) legumes (+) fish (+) red and processed meat (−) dairy (+) USFA:SFA (+)	Median intake (0,1) ^¶^	0–9	No association shown between MDS and recurrent wheeze, asthma, eczema, or lower respiratory infection in offspring at 3 years.
Lindsay et al. (2020) [33]	Cohort study, USA	203 mothers	Repeated 24-hour recalls, T1, T2, and T3	MDS-2003 [20]	9 components: vegetables (+) fruits (+) nuts (+) whole grains (+) legumes (+) fish and shellfish (+) red and processed meat (−) dairy (+) MUFA:SFA (+)	Fixed criteria (0,1) ^§^	0–9	No association shown between MDS and maternal HOMA-IR.
Mantzoros et al. (2010) [34]	Cohort study (Viva), USA	780 mother-child pairs	Semi-quantitative FFQ, T1 and T2	MDS-2003 [13,52]	9 components: vegetables (+) fruits (+) nuts (+) whole grains (+) legumes (+) fish (+) red and processed meat (−) dairy (+) USFA:SFA (+)	Median intake (0,1) ^¶^	0–9	No association shown between MDS and cord-blood leptin or adiponectin.
Martínez-Galiano et al. (2018) [23]	Case-control study, Spain	1036 mother-child pairs	Semi-quantitative, 137-item FFQ, post-partum	MDS-2003 [13]; PREDIMED score [53]; MDS-2006 [54]	8 components (MDS-2003): vegetables (.) fruits (.) cereals (.) legumes (.) fish (.) meat (.) dairy (.) MUFA:SFA (.) 12 components (PREDIMED):^†††^ vegetables (+) fruits (+) nuts (+) legumes (+) fish (+) chicken, turkey, or rabbit (+) red meat, hamburger, or sausages (−) butter, margarine, or cream (−) olive oil (+) commercial sweets or pastries (−) dishes with sofrito (+)^§§§^ sweet or carbonated beverages (−) 10 components (MDS-2006): vegetables (+) potatoes (+) fruits (+) whole grains (+) legumes (+) fish (+) meat (−) poultry (−) dairy with fat (−) olive oil (+)	MDS-2003 – median intake (0,1) ^¶^ PREDIMED – fixed criteria (0,1) ^¶¶¶^ MDS-2006 – fixed criteria (0–5)^§^	MDS-2003 (0–8); PREDIMED (0–13); MDS-2006 (0–50)	Inverse association shown between MDS-2003 and SGA. No association shown between PREDIMED score and MDS-2006 and SGA.
Mikkelsen et al. (2008) [35]	Cohort study (DNBC), Denmark	35,530 mother-child pairs	Semi-quantitative, 360-item FFQ, T2	Khoury’s criteria [51]	5 components: vegetables and fruits (+) fish (+) meat (−) olive oil or rapeseed oil (+) coffee (−)	Fixed criteria (0,1) ^¶¶¶^	0–5	Inverse association shown between MDS and early PTB.
Peraita-Costa et al. (2018) [42]	Cross-sectional study, Spain	492 mothers	16-question KIDMED questionnaire, post-partum	Modified KidMed index [55]	13 components:^††††^ vegetables (+) fruits (+) nuts (+) cereals (+) pasta or rice (+) legumes (+) fish (+) dairy (+) olive oil (+) fast foods (−) commercial baked goods or pastries (−) sweets and candy (−) skipping breakfast (−)	Fixed criteria (−1,+1) ^¶¶¶^	0–12	No association shown with SGA.
Peraita-Costa et al. (2020) [46]	Nested case-control study, Spain	1118 mothers	16-question KIDMED questionnaire, post-partum	Modified KidMed index [55]	13 components: ^††††^ vegetables (+) fruits (+) nuts (+) cereals (+) pasta or rice (+) legumes (+) fish (+) dairy (+) olive oil (+) fast foods (−) commercial baked goods or pastries (−) sweets and candy (−) skipping breakfast (−)	Fixed criteria (−1,+1) ^¶¶¶^	0–12	Moderate MDS shown to be associated with increased risk of PTB. No association between MDS and SGA.
Poon et al. (2013) [36]	Population-based cohort (IFPSII), USA	893 mother-infant pairs	FFQ, T3	aMed [13]	8 components: vegetables (+) fruits (+) nuts (+) whole grains (+) legumes (+) fish (+) red and processed meat (−) MUFA:SFA (+)	Median intake (0,1) ^¶^	0–8	No association shown between MDS and LGA, SGA, birth weight, or WFL at birth, or change in WFL from 4–6 months.
Saunders et al. (2014) [37]	Population-based cohort study (TIMOUN), Caribbean	728 mother-child pairs	Semi-quantitative, 214-item FFQ, post-partum	MDS-2003 [12,56]	9 components: vegetables (+) fruits and nuts (+) cereals (+) legumes (+) fish (+) meat (−) dairy (+) alcohol (−) MUFA:SFA (.)	Median intake (0,1) ^§§^	0–9	No association shown between MDS and PTB and FGR.
Spadafranca et al. (2014) [43]	Cross-sectional study, Italy	99 mothers	FFQ, T1 and T3	PREDIMED score [57]	12 components: ^†††^ vegetables (+) fruits (+) nuts (+) legumes (+) fish and shellfish (+) chicken, turkey, or rabbit (+) red meat, hamburger, or sausages (−) butter, margarine, or cream (−) olive oil (+) commercial sweets or pastries (−) dishes with sofrito (+) ^§§§^ sweet or carbonated beverages (−)	Fixed criteria (0,1) ^¶¶¶^	0–13	Inverse association shown between MDS and level of adiponectin in mothers from T1 to T3.
Tomaino et al. (2020) [44]	Cross-sectional study, Spain	218 mother-child pairs	14-question, MEDAS questionnaire, post-partum	PREDIMED score [50,58]	13 components:^†††^ vegetables (+) fruits (+) nuts (+) legumes (+) fish and shellfish (+) chicken, turkey, or rabbit (+) red meat, hamburger, or sausages (−) butter, margarine, or cream (−) olive oil (+) wine (+) commercial sweets or pastries (−) dishes with sofrito (+) ^§§§^ sweet or carbonated beverages (−)	Fixed criteria (0,1) ^¶¶¶^	0–14	No association shown between MDS and infant weight at birth.

* Dietary components with (+) indicates contributed to increased adherence to the MD; (−) indicates contributed to decreased adherence to the MD; (.) indicates contribution was unspecified within the study’s methods. **^†^** Lowest number in range indicates minimal adherence to the MD and highest number indicates maximal adherence. ^§^ Cut-off points based on fixed servings per day or week. ^¶^ Women with higher intakes of beneficial foods greater than the median intake received +1; women with lower intakes of beneficial foods less than the median received 0. ** The number of pregnant women or mothers was not specified within the study. **^††^** Unspecified time period during pregnancy in which dietary data was collected. ^§§^ Women with higher intakes of beneficial foods greater than or equal to the median intake received +1; women with lower intakes of beneficial foods less than the median received 0. ^¶¶^ Women with upper half of intake of beneficial foods received +1; women with lower half of intake of detrimental foods received +1. *** No information provided on components, cut-off points, or scoring system of aMed. **^†††^** PREDIMED score includes two questions on olive oil and one question on every other component. ^§§§^ Sofrito was described as a sauce consisting of tomatoes, garlic, onion, and peppers or leeks sautéed in olive oil and served with dishes of vegetables, rice, or pasta. ^¶¶¶^ Cut-off points based on combination of fixed servings per week or month and food habits. **** Alcohol was scored dichotomously: 0 for any consumption and 2 for no consumption. **^††††^** KIDMED questionnaire includes two questions each on vegetables, fruits, and dairy products and one question on every other component.

**Table 3 nutrients-13-00582-t003:** Comparison of food and nutrient components *, cut-off points, and range of scores in *a priori* indexes assessing maternal adherence to the MD during pregnancy.

	Vegetables ^†^		Fruits and Nuts ^§^			Cereals ^¶^			Legumes	Fish **		Meat ^††^			Dairy ^§§^				Alcohol	Lipid Ratio		Other Components							Cut-off points ^¶¶^		Range of Scores ***
	Vegetables	Potatoes	Fruits and Nuts	Fruits	Nuts	Cereals	Pasta and Rice	Non-refined or Whole Grains	Legumes	Fish	Fish and Shellfish or Seafood	Meat	Poultry	Red and Processed	Dairy	Full-Fat Dairy or Dairy with Fat	Milk	Butter, Margarine, or Cream ^†††^	Alcohol	MUFA:SFA	USFA:SFA	Olive Oil or Rapeseed Oil	Fast Food and Junk Foods	Sweets, Candies, and Pastries	Dishes with Sofrito^§§§^	Sweetened or Carbonated Beverages	Coffee	Skipping Breakfast	Distribution	Fixed Criteria	
Babili et al. (2020) [38]																														×	0–55
Bédard et al. (2020) [24]																													×		0–7
Castro-Rodriguez et al. (2010) [25] ^¶¶¶^ ****																														×	0–22
Castro-Rodriguez et al. (2016) [26] ^¶¶¶^ ****																														×	0–22
Chatzi et al. (2008) [27]																													×		0–7
Chatzi et al. (2012) [28]																													×		0–8
Chatzi et al. (2013) [29]																													×		0–8
Chatzi et al. (2017) [20]																														×	0–9
de Batlle et al. (2008) [39] ^¶¶¶^																													×		0–8
Fernández-Barrés et al. (2016) [21]																													×		0–16
Fernández-Barrés et al. (2019) [30]																													×		0–16
Gesteiro et al. (2012) [40] ^††††^																														×	0–13
Gesteiro et al. (2015) [41] ^††††^																														×	0–13
Haugen et al. (2008) [31] ^§§§§^																														×	0–5
Izadi et al. (2016) [45]																													×		0–9
Jardí et al. (2019) [22]																													×		0–18
Lange et al. (2010) [32]																													×		0–9
Lindsay et al. (2020) [33]																														×	0–9
Mantzoros et al. (2010) [34]																													×		0–9
Martínez-Galiano et al. (2018) [23] ^¶¶¶¶^																															
MDS-2003																													×		0–8
PREDIMED score																														×	0–13
MDS-2006																														×	0–50
Mikkelsen et al. (2008) [35] ^§§§§^																														×	0–5
Peraita-Costa et al. (2018) [42]																														×	0–12
Peraita-Costa et al. (2020) [46]																														×	0–12
Poon et al. (2013) [36]																													×		0–8
Saunders et al. (2014) [37]																													×		0–9
Spadafranca et al. (2014) [43] ^††††^																														×	0–13
Tomaino et al. (2020) [44] ^††††^																														×	0–14

* Color of boxes indicates contribution to MDS: Green (increased adherence to MD); red (decreased adherence to MD); yellow (unspecified contribution); and white (exclusion of component from index). Number of boxes may not add up to exact number of components included in index due to differences in categorization of some components. **^†^** Vegetables category compares studies including a single vegetable component and those including a separate component for potatoes. ^§^ Fruits and nuts category compares studies including fruits and nuts as a single component and those including fruits and nuts as separate components. ^¶^ Cereals category compares studies including cereals as a single component; those including separate components for pasta and/or rice; and, those specifying only the inclusion of non-refined or whole grains. ****** Fish category compares studies including fish and those additionally including shellfish or seafood in the fish component. **^††^** Meat category compares studies including meat as a single component; those including poultry as a separate category; and those specifying only the inclusion of red and processed meat. ^§§^ Dairy category compares studies including dairy as a single component; those including only full-fat dairy or dairy with fat; those including only milk; and those including only butter, margarine, or cream. ^¶¶^ (×) indicates cut-off points based on distribution or fixed criteria. ^***^ Lowest number indicates minimal adherence to the MD and highest number indicates maximal adherence to MD. **^†††^** Non-dairy component of margarine grouped with butter and cream in the studies utilizing the PREDIMED score. ^§§§^ Sofrito was described as a sauce consisting of tomatoes, garlic, onion, and peppers or leeks sautéed in olive oil and served dishes of vegetables, pasta, or rice. ^¶¶¶^ Studies utilized a generic category for fast food also encompassing other foods (i.e., sweets, pastries, snacks, and fat). **** Studies separated pasta and rice into separate categories. **^††††^** Studies scored group rabbit with poultry components (i.e., chicken and turkey). ^§§§§^ Studies grouped vegetables and fruit together into a single category. ^¶¶¶¶^ Martínez-Galiano et al. (2018) utilized three different indexes: MDS-2003, PREDIMED score, and MDS-2006.

## Data Availability

Not applicable.

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
