# Peer review of "Maternal Adherence to the Mediterranean Diet during Pregnancy: A Review of Commonly Used a priori Indexes"

_nutrients, 2021, doi:10.3390/nu13020582_

Round 1
Reviewer 1 Report
Review of a manuscript: “Maternal adherence to the Mediterranean duet during pregnancy: a review of commonly used a priori indexes"
This study is about the assessment of adherence to Mediterranean diet during pregnancy using a priori indexes. As mentioned adherence to the Mediterranean diet is widely studied in relation to health outcomes. However, several versions of indexes to assess the adherence vary between studies and even researchers modified the score by the year it has been in use. In regard to cutoff, some studies used median intake while other used fixed criteria. Beyond the vast heterogeneity among indexes, assessing adherence to Mediterranean diet during pregnancy further complicate the use of Mediterranean indexes among pregnant women. The current review focuses on this population. This information that the author review was done using a well organized approach. However, the main objective of the current paper is not clear. The differences in indexes are due to the inherent difference in dietary habits; the same is true for the definitions of cutoffs. For example the authors mention Sofritos in the Spanish index, a food that is probably consumed in Spain but not in other countries. Thus the same score will need to be modified for use in other countries.
Therefore the use of various indexes to measure adherence to Mediterranean diet is essential, especially among pregnant women, with their specific needs and different eating patterns. Each dietary study among pregnant women will for sure need to modify questionnaire based on the local eating patterns of the population. It may limit the ability to compare between studies although some core components of the MD are similar across studies. [olive oil, vegetables and fruits, grains and legumes].
Based on my review, my recommendation to the author is to include their findings in a paper that describes the development/adaptation and validation of MD adherence score for a population that they plan to study. Otherwise, this review does not stand by its own.
With regard to the manuscript:
It is well and clear written manuscript.
Author Response
We would like to thank you for your time and expertise in evaluating our manuscript “Maternal adherence to the Mediterranean diet during pregnancy: a review of commonly used a priori indexes” for publication in Nutrients. With this letter, we have aimed to address each of the reviewer’s comments and have indicated changes to the manuscript as applicable. Below, you will find a point-by-point address to the reviewers’ comments.
Comment 1: “However, the main objective of the current paper is not clear. The differences in indexes are due to the inherent difference in dietary habits; the same is true for the definitions of cutoffs. For example the authors mention Sofritos in the Spanish index, a food that is probably consumed in Spain but not in other countries. Thus the same score will need to be modified for use in other countries. Therefore the use of various indexes to measure adherence to Mediterranean diet is essential, especially among pregnant women, with their specific needs and different eating patterns. Each dietary study among pregnant women will for sure need to modify questionnaire based on the local eating patterns of the population. It may limit the ability to compare between studies although some core components of the MD are similar across studies. [olive oil, vegetables and fruits, grains and legumes].”
Reply: The main objective of the manuscript was “to identify all observational studies utilizing a priori indexes to assess maternal adherence to the Mediterranean diet during pregnancy with a particular emphasis on evaluating the food and nutrients components, namely the choice and discrimination of dietary components in these indexes, in addition to the cut-off values and scoring systems” (Page 2 Lines 83-87). Findings from this review illustrate the vast heterogeneity of these a priori indexes, which are not necessarily or solely due to inherent differences in local eating habits of different cultures. For instance, Table 1 of this review shows that Castro-Rodriguez et al. (2010) and Chatzi et al. (2008) are both conducted in Spain but do not list sofrito as a separate dietary component in their indexes. Thus, comparability among studies is hampered even when conducted within pregnant populations of the same culture. Even more so, an important finding from this review is that only three components—vegetables, fruits, and fish—were common to all indexes of the twenty-seven included studies. Therefore, not all components largely considered traditional to the Mediterranean diet (i.e. legumes and cereals) were incorporated into the indexes. This presents a further discrepancy among indexes, which goes beyond any modifications of questionnaires to local eating patterns.
Comment 2: “Based on my review, my recommendation to the author is to include their findings in a paper that describes the development/adaptation and validation of MD adherence score for a population that they plan to study. Otherwise, this review does not stand by its own.”
Reply: Thank you for this interesting suggestion. While we acknowledge the relevance of validating an a priori index for use in a specific population, this suggestion is beyond the scope of this review. The objective of this review was to identify all observational studies utilizing a priori indexes used to assess maternal adherence to the Mediterranean diet during pregnancy with a particular emphasis on evaluating the inclusion and discrimination of dietary components as well as the cut-off values and scoring systems. We believe that this review provides a
necessary overview of variations of a priori indexes used to assess adherence to the Mediterranean diet in pregnant populations. Firstly, this will assist future studies in both realizing the vast heterogeneity among a priori indexes, which greatly complicates comparability among studies even when evaluating the same outcome measure. Secondly, it will assist future studies in choosing the most suitable index to assess maternal adherence to the Mediterranean diet during pregnancy for their specific research question prior to application in a specific pregnant population, including considerations for modifying indexes for pregnancy-specific nutritional concerns.
Comment 3: “With regard to the manuscript: It is well and clear written manuscript.”
Reply: Thank you for your kind words.
In conclusion, we would like to reiterate our sincere gratitude for evaluating our manuscript. This invaluable opportunity has allowed us to revise and further improve our manuscript. We believe that this manuscript could greatly assist future studies in choosing the most suitable index to assess maternal adherence to the Mediterranean diet during pregnancy for their research. As such, we sincerely hope that we have sufficiently addressed each of the reviewer’s comments. If necessary, we would be amenable to revising the manuscript based on additional comments.
We look forward to hearing from you soon.
On behalf of all authors
Reviewer 2 Report
Manuscript “Maternal adherence to the Mediterranean diet during pregnancy: a review of commonly used a priori indexes” (Nutrients 1060550).
This narrative review aimed to identify all observational studies utilizing a priori indexes to assess maternal adherence to the Mediterranean diet (MD) during pregnancy. The authors, after analyzing their results, conclude that "this review emphasizes the incongruent definitions of the MD impairing effective comparison among studies relating to maternal or offspring health outcomes. Future research should carefully consider the heterogeneous definitions of the MD in a priori indexes and the relevance of incorporating pregnancy-specific nutritional requirements”. I think this is a very solid conclusion and should be taken into account.
Comments and Suggestions for Authors:
The manuscript is a very interesting narrative review, but requires some minor considerations.
Page 1 Line 14, Page 11 Line 194, Page 17 Line 467. The sentences "The number of dietary components ranged from five to thirteen. Of the seventeen dietary components identified ..." can be confusing at first and could be explained further.
Page 2 Line 90. It should be specified if custom range has been used, with the start date.
Page 18 Line 497 and 521. The reference to the entity type II diabetes mellitus should be better done with Arabic numerals (type 2 diabetes mellitus).
- Aune D, Norat T, Romundstad P, Vatten LJ. Whole grain and refined grain consumption and the risk of type 2 diabetes: a systematic review and dose-response meta-analysis of cohort studies. Eur J Epidemiol. 2013;28(11):845–58.
- American Diabetes Association. 2. Classification and Diagnosis of Diabetes: Standards of Medical Care in Diabetes-2020. Diabetes Care. 2020;43(Suppl 1): S14-S31.
References should be thoroughly revised to conform to uniform and appropriate standards for the journal Nutrients.
Author Response
We would like to thank you for your time and expertise in evaluating our manuscript “Maternal adherence to the Mediterranean diet during pregnancy: a review of commonly used a priori indexes” for publication in Nutrients. With this letter, we have aimed to address each of the reviewer’s comments and have indicated changes to the manuscript as applicable. Below, you will find a point-by-point address to the reviewers’ comments:
Comment 1: “ ‘This narrative review aimed to identify all observational studies utilizing a priori indexes to assess maternal adherence to the Mediterranean diet (MD) during pregnancy. The authors, after analyzing their results, conclude that "this review emphasizes the incongruent definitions of the MD impairing effective comparison among studies relating to maternal or offspring health outcomes. Future research should carefully consider the heterogeneous definitions of the MD in a priori indexes and the relevance of incorporating pregnancy-specific
nutritional requirements’ ”. I think this is a very solid conclusion and should be taken into account.”
Reply: Thank you for your kind words.
Comment 2: “Page 1 Line 14, Page 11 Line 194, Page 17 Line 467. The sentences ‘The number of dietary components ranged from five to thirteen. Of the seventeen dietary components identified ...’ can be confusing at first and could be explained further.”
Reply: We have amended these sentences in order to better avoid potential confusion for readers, while still adhering to the maximum word count (200 words) for the abstract.
§ Page 1 Line 14-17: “Studies included a range of five to thirteen dietary components in their indexes. Only three dietary components—vegetables, fruits, and fish—were common among all indexes.”
§ Page 12 Line 195-201 (formerly Page 11 Line 194): “Overall, the number of dietary components included in the indexes ranged from five [31,35] to thirteen [42,44,46] depending upon the chosen index and any subsequent modifications to that index. In total, however, seventeen different dietary components were
identified among the indexes of the twenty-seven studies. Ten components were considered traditional to the components included in the original MDS and MDS-2003 [12,13]; seven components were considered non-traditional to these original indexes [12,13].”
§ Page 18 Line 478-483 (formerly Page 17 Line 467): “Studies included a range of five [31,35] to thirteen [42,44,46] dietary components in their indexes; however, a total of seventeen different dietary components were identified among the indexes of the twenty-seven studies. Ten components were considered traditional to the original indexes of the MD [12,13].”
Comment 3: “Page 2 Line 90. It should be specified if custom range has been used, with the start date.”
Reply: We have specified the lack of time limits used in the database search for relevant literature.
§ Page 2 Line 89-91 (formerly Page 2 Line 90): “A comprehensive search was conducted in Pubmed in order to identify all observational studies published evaluating maternal adherence to the MD during pregnancy without time limits through July 1, 2020.”
Comment 4: “Page 18 Line 497 and 521. The reference to the entity type II diabetes mellitus should be better done with Arabic numerals (type 2 diabetes mellitus).”
Reply: We have implemented the Arabic numeral “2” instead of “II” in “type II diabetes mellitus” in all applicable portions of the manuscript.
§ Page 19 Line 510 (formerly Page 18 Line 497)
§ Page 19 Line 534 (formerly Page 18 521)
§ List of abbreviations on Page 22
Comment 5: “References should be thoroughly revised to conform to uniform and appropriate standards for the journal Nutrients.”
Reply: We have revised the in-text references and reference list to meet the standards of Nutrients.
In conclusion, we would like to reiterate our sincere gratitude for evaluating our manuscript. This invaluable opportunity has allowed us to revise and further improve our manuscript. We believe that this manuscript could greatly assist future studies in choosing the most suitable index to assess maternal adherence to the Mediterranean diet during pregnancy for their research. As such, we sincerely hope that we have sufficiently addressed each of the reviewer’s comments. If necessary, we would be amenable to revising the manuscript based on additional comments.
We look forward to hearing from you soon.
On behalf of all authors
Reviewer 3 Report
line 89-90 - what period of time did the research come from? what was the bottom date?
lines 119-120 should be transferred to Materials and Methods
lines 149-150 -maternal adherence to the Mediterranean diet in relation to maternal health outcomes: including the risk of GDM.- There are more publications looking for links between MD and GDM in pregnant women, should these works not be used for the presented literature review?
Schoenaker D.A., Soedamah-Muthu S.S., Callaway L.K., Mishra G.D. Pre-pregnancy dietary patterns and risk of gestational diabetes mellitus: Results from an Australian population-based prospective cohort study. Diabetologia. 2015;58:2726–2735. doi: 10.1007/s00125-015-3742-1.
Donazar-Ezcurra M., Lopez-Del Burgo C., Martinez-Gonzalez M.A., Basterra-Gortari F.J, de Irala J., Bes-Rastrollo M. Pre-pregnancy adherences to empirically derived dietary patterns and gestational diabetes risk in a Mediterranean cohort: The Seguimiento Universidad de Navarra (SUN) project. Br. J. Nutr. 2017;118:715–721. doi: 10.1017/S0007114517002537.
Olmedo-Requena R, Gómez-Fernández J, Amezcua-Prieto C, Mozas-Moreno J, Khan KS, Jiménez-Moleón JJ. Pre-Pregnancy Adherence to the Mediterranean Diet and Gestational Diabetes Mellitus: A Case-Control Study. Nutrients. 2019 May 1;11(5):1003. doi: 10.3390/nu11051003. PMID: 31052474; PMCID: PMC6566892.
Maternal dietary patterns and gestational diabetes mellitus: a large prospective cohort study in China.
He JR, Yuan MY, Chen NN, Lu JH, Hu CY, Mai WB, Zhang RF, Pan YH, Qiu L, Wu YF, Xiao WQ, Liu Y, Xia HM, Qiu X
Br J Nutr. 2015 Apr 28; 113(8):1292-300.
Presented review emphasizes the vast heterogeneity and subjectiity present among the components, cut-off points, and scoring systems of a priori indexes used to assess maternal adherence to the MD during pregnancy. The study is a very valuable collection of knowledge on the impact of the Mediterranean diet on the health of pregnant women and shows the share of specific recommended and non-recommended groups of products in the diet of pregnant women.
Author Response
We would like to thank you for your time and expertise in evaluating our manuscript “Maternal adherence to the Mediterranean diet during pregnancy: a review of commonly used a priori indexes” for publication in Nutrients. With
this letter, we have aimed to address each of the reviewer’s comments and have indicated changes to the manuscript as applicable. Below, you will find a point-by-point address to the reviewers’ comments:
Comment 1: “line 89-90 - what period of time did the research come from? what was the bottom date?”
Reply: We have specified the lack of time limits used in the database search for relevant literature.
§ Page 2 Line 89-91 (formerly Page 2 Line 90): “A comprehensive search was conducted in Pubmed in order to identify all observational studies published evaluating maternal adherence to the MD during pregnancy without time limits through July 1, 2020.”
Comment 2: “lines 119-120 should be transferred to Materials and Methods”
Reply: Thank you for your suggestion to move Page 6 Lines 120-121 (formerly Page 5 Lines 119-120) to the “Materials and Methods” section: “The studies were published from 2008 to 2020.” We prefer to keep this sentence in the “Results” section to avoid any potential confusion with readers given that we did not utilize time constraints in our search, which hopefully has been clarified in the aforementioned revision. We identified studies published from 2008 to 2020; however, we did not exclude any studies based solely on being published prior to 2008.
Comment 3: “lines 149-150 maternal adherence to the Mediterranean diet in relation to maternal health outcomes: including the risk of GDM. There are more publications looking for links between MD and GDM in pregnant women, should these works not be used for the presented literature review? Schoenaker D.A., Soedamah-Muthu S.S., Callaway L.K., Mishra G.D. Pre-pregnancy dietary patterns and risk of gestational diabetes mellitus: Results from an Australian population-based prospective cohort study. Diabetologia. 2015;58:2726–2735. doi: 10.1007/s00125-015-3742-1. Donazar-Ezcurra M., Lopez-Del Burgo C., Martinez-Gonzalez M.A., Basterra-Gortari F.J, de Irala J.,Bes-Rastrollo M. Pre-pregnancy adherences to empirically derived dietary patterns and gestational diabetes risk
in a Mediterranean cohort: The Seguimiento Universidad de Navarra (SUN) project. Br. J. Nutr. 2017;118:715–721. doi: 10.1017/S0007114517002537. Olmedo-Requena R, Gómez-Fernández J, Amezcua-Prieto C, Mozas-Moreno J, Khan KS, Jiménez-Moleón JJ. Pre-Pregnancy Adherence to the Mediterranean Diet and Gestational Diabetes Mellitus: A Case-Control Study. Nutrients. 2019 May 1;11(5):1003. doi: 10.3390/nu11051003. PMID: 31052474; PMCID: PMC6566892. Maternal dietary patterns and gestational diabetes mellitus: a large prospective cohort study in China. He JR, Yuan MY, Chen NN, Lu JH, Hu CY, Mai WB, Zhang RF, Pan YH, Qiu L, Wu YF, Xiao WQ, Liu Y, Xia HM, Qiu X Br J Nutr. 2015 Apr 28; 113(8):1292-300.”
Reply: After applying exclusion criteria to the studies identified in our literature search, we identified only one study—Izadi et al. (2016)—that assessed gestational diabetes mellitus as an outcome measure as indicated in Page 11 Line 151 (formerly Page 10 Line 150). Of the studies listed above, Schoenaker et al. (2015), Donazar- Ezcurra et al. (2017), and Olmedo-Requena et al. (2019) were excluded as they assessed pre-pregnancy maternal adherence to the Mediterranean diet rather than specifically assessing maternal adherence to the Mediterranean diet during pregnancy. He et al. (2015) was not included as they utilized principal components factor analysis rather than an a priori indexes to assess maternal adherence to the Mediterranean diet during pregnancy. We have
detailed the exclusion criteria in the legend of Figure 1, including clarifying the exclusion of studies assessing prepregnancy study populations.
Comment 4: “The study is a very valuable collection of knowledge on the impact of the Mediterranean diet on the health of pregnant women and shows the share of specific recommended and non-recommended groups of products in the diet of pregnant women.”
Reply: Thank you for your kind words.
In conclusion, we would like to reiterate our sincere gratitude for evaluating our manuscript. This invaluable opportunity has allowed us to revise and further improve our manuscript. We believe that this manuscript could greatly assist future studies in choosing the most suitable index to assess maternal adherence to the Mediterranean diet during pregnancy for their research. As such, we sincerely hope that we have sufficiently addressed each of the reviewer’s comments. If necessary, we would be amenable to revising the manuscript based on additional comments.
We look forward to hearing from you soon.
On behalf of all authors
Marion
Round 2
Reviewer 1 Report
No more comments, the manuscript was modified according to review suggestions